# Effects of Enhancement on Deep Learning Based Hepatic Vessel Segmentation

**Shanmugapriya Survarachakan** [1], **Egidijius Pelanis** [2,3,†], **Zohaib Amjad Khan** [4,†], **Rahul Prasanna Kumar** [2], **Bjørn Edwin** [2,3,5] **and Frank Lindseth** [1,*]

1   Department of Computer Science, Norwegian University of Science and Technology,
    7491 Trondheim, Norway; shanmugapriya.survarachakan@ntnu.no
2   The Intervention Centre, Oslo University Hospital Rikshospitalet, Pb. 4950 Nydalen, 0424 Oslo, Norway;
    egidijus@pelanis.eu (E.P.); rahul.kumar@ous-research.no (R.P.K.); bjoedw@ous-hf.no (B.E.)
3   Institute of Clinical Medicine, Faculty of Medicine, University of Oslo, 0315 Oslo, Norway
4   L2TI, Institut Galilée, Université Sorbonne Paris Nord, UR 3043, 93430 Villetaneuse, France;
    zohaibamjad.khan@univ-paris13.fr
5   Department of HPB Surgery, Oslo University Hospital Rikshospitalet, 0372 Oslo, Norway
*   Correspondence: frankl@ntnu.no
†   These authors contributed equally to this work.

**Abstract:** Colorectal cancer (CRC) is the third most common type of cancer with the liver being the most common site for cancer spread. A precise understanding of patient liver anatomy and pathology, as well as surgical planning based on that, plays a critical role in the treatment process. In some cases, surgeons request a 3D reconstruction, which requires a thorough analysis of the available images to be converted into 3D models of relevant objects through a segmentation process. Liver vessel segmentation is challenging due to the large variations in size and directions of the vessel structures as well as difficult contrasting conditions. In recent years, deep learning-based methods had been outperforming the conventional image analysis methods in the field of medical imaging. Though Convolutional Neural Networks (CNN) have been proved to be efficient for the task of medical image segmentation, the way of handling the image data and the preprocessing techniques play an important role in segmentation. Our work focuses on the combination of different vesselness enhancement filters and preprocessing methods to enhance the hepatic vessels prior to segmentation. In the first experiment, the effect of enhancement using individual vesselness filters was studied. In the second experiment, the effect of gamma correction on vesselness filters was studied. Lastly, the effect of fused vesselness filters over individual filters was studied. The methods were evaluated on clinical CT data. The quantitative analysis of the results in terms of different evaluation metrics from experiments can be summed up as (i) each of the filtered methods shows an improvement as compared to unenhanced with the best mean DICE score of 0.800 in comparison to 0.740 for unenhanced; (ii) applied gamma correction provides a statistically significant improvement in the performance of each filter with improvement in mean DICE of around 2%; (iii) both the fused filtered images and fused segmentation give the best results (mean DICE score of 0.818 and 0.830, respectively) with the statistically significant improvement compared to the individual filters with and without Gamma correction. The results have further been verified by qualitative analysis and hence show the importance of our proposed fused filter and segmentation approaches.

**Keywords:** hepatic vessels; enhancement; Frangi; Hessian; Meijering; Sato; gamma correction; CT

## 1. Introduction

Colorectal cancer is the third most common type of cancer with ≈1.9 million new cases and ≈935 thousand deaths yearly [1]. At the time of diagnosis, approximately a quarter of the patients already have cancer spread into the liver in the form of metastasis [2,3]. Detection, definition and mapping of these metastases require medical imaging.

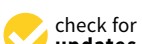



Medical image segmentation is the process of locating and extracting the anatomical structures of interest or pathologies from medical images such as computed tomography, magnetic resonance imaging, ultrasound, etc. Extraction of hepatic vessels and their relationship with tumors play an important role in liver surgery treatment and planning [4]. In addition, liver vessel extraction aids in visualization, liver segment approximation, multi-modal registration, where vessels act as landmarks, computer-aided diagnosis and surgery [4,5]. The manual delineation of hepatic vessels is often time-consuming, error-prone and highly user dependent. Image-related challenges, such as high signal-to-noise ratio, low image resolution, inhomogeneous background, varying contrast between the vessels and liver parenchyma, imaging artifacts and varying vessel thickness [6] make hepatic vessel segmentation challenging. In recent years, deep learning-based methods have become widely used for various medical image analysis tasks. Though deep learning-based segmentation is proven to be efficient, the way of data handling and the enhancement of images that go into the deep learning model has a major influence on the precise segmentation. Especially for complex structures such as vessels, enhancement techniques are proven to be effective prior to segmentation and visualization [7–9]. In particular, Hessian-based vessel enhancement filters are most popularly used compared to other techniques [10–14].

In this work, the impact of four different multi-scale vesselness filters, including Hessian, Sato, Frangi and Meijering, were studied experimentally. The effect of enhancement has been evaluated based on 3D U-net, a widely used deep-learning based segmentation model in the medical imaging domain. To review the impact of enhancement on segmentation, we compared the segmentation results with and without vesselness enhancement. We also compared the performance of different multi-scale vesselness filter. Secondly, we studied the effect of gamma filtering on vesselness enhanced images. Finally, we proposed to fuse the outcome from the filtered and gamma corrected images in two different designs, and the effect of fusing the outcome is compared over the individual vesselness filter. We evaluated the methods on the clinical dataset.

## 2. Related Work

In Frangi et al. [10], the vesselness measure is obtained based on all eigenvalues of the Hessian, a multi-scale second order local structure of the image. Yang et al. [12] improved the multi-scale enhancement technique inspired by Frangi et al. [10] to enhance the vessel structures for retinal vessel segmentation. Kumar et al. [15] developed a modified multi-vesselness filter based on the Hessian matrix for the center line extraction of blood vessels. In Drechsler et al. [13], the enhancement method is based on the Laplacian for the tubular structures. Jerman et al. [14] implemented a enhancement filter based on the ratio of multi-scale Hessian eigenvalues to accurately enhance the borders between the vessel structures and the background. In Lamy et al. [16], seven different Hessian based vesselness filters were compared and benchmarked on the IRCAD and VasucSynth datasets. The enhancement methods were evaluated using level-set-based segmentation. Zeng et al. [17] proposed a Hessian-based multi-feature method to segment the liver vessel structures prior to segmentation based on the extreme learning machine. Manh Luu et al. [18] evaluated five different diffusion filters for enhancing the liver vessels in 3D CTA images. In Phellan and Forkert [19], different vessel enhancement algorithms were applied to time-of-flight MRA images for cerebrovascular segmentation and then compared.

In Shahid and Taj [7], a combination of adaptive histogram equalization, morphological top-hat filter, high boost filtering and Frangi filter were employed for the purpose of retinal vessel enhancement prior to deep learning-based segmentation. In Soomro et al. [8], morphological operators and the contrast limited adaptive histogram equalization (CLAHE) technique were used to enhance and segment the retinal vessels. In Blaiech et al. [9], to study the effect of enhancement, CLAHE, Frangi and ranking the orientation response

of path operators(RORPO) were used to enhance the coronary artery for the purpose of segmentation.

Gamma correction is widely used in the medical image enhancement process. In Karuppanagounder and Palanisamy [20], a gamma correction-based technique is used to enhance medical MRI and CT images. In Tiwari and Gupta [21], a combination of gamma correction and homomorphic filtering was used to enhance knee MRI images. In Dash and Senapati [22] and Zhitao et al. [23], the combination of gamma correction with other enhancement techniques was used to enhance the retinal blood vessels from the ophthalmic images.

In the field of medical image segmentation, 3D U-net [24] based architectures are widely used for various volumetric segmentation tasks [25].

Inspired from the literary works, in this paper, we also studied the effect of the combination of Hessian-based vesselness filters and gamma-correction technique for the hepatic vessel enhancement. We also proposed fusing the outcome from the vesselness filtered and gamma-corrected images to improve the effect of enhancement for the purpose of deep learning-based hepatic vessel segmentation.

This paper consists of seven sections. The Section 1 gives an introduction to the paper and the motivation behind it. The Section 2 describes the background and related work. The various methods and the datasets used in the paper are presented in Section 3. The experiments comparing different methods are presented in Section 4 and their corresponding results are presented in Section 5. Section 6 summarizes the discussion of the results and Section 7 presents the conclusion and suggests future work.

## 3. Materials and Methods

In this section, the dataset, experimental setup, methodology and different evaluation techniques used in the experiments are explained.

### 3.1. Dataset

This study utilizes a dataset derived from Oslo University hospital OSLO-COMET (Oslo Randomized Laparoscopic Versus Open Liver Resection for Colorectal Metastases Trial). (ClinicalTrials.gov: NCT01516710) with necessary approvals from both local and regional ethical committees. All study participants had colorectal liver metastasis and were referred for liver resection. The used dataset consists of 57 contrast-enhanced computed tomography (CT) images from 4 manufacturers and 13 different CT machine models. These volumes were processed and manually segmented by a medical doctor. The segmentation process involved a combination of ITK-SNAP [26] and SLICER [27] to annotate liver parenchyma and clearly visible vessels. All datasets were scaled to an isotropic resolution of $1 \times 1 \times 1$ mm. We split the dataset as 70% for training, 20% for validation and 10% for testing.

### 3.2. Experimental Setup

The vessel enhancement and gamma-correction filters were implemented using the scikit-image library [28] in python. For segmentation, the Chainer implementation of 3D U-net [24] was used. In total, 11 models based on different inputs (Unenhanced, Frangi, Hessian, Meijering, Sato, FrangiGC, HessianGC, MeijeringGC, SatoGC, FilterAdded) were trained to segment the hepatic vessel structures. The models were trained on a cluster with multiple V100 GPUs with 32GB video memory. All models were trained using the same hyper-parameter configurations including softmax-cross-entropy loss, adam optimizer, ReLU activation function and initial learning rate of 0.0001. The weights were initialized using He's initialization method. A patch size of $64 \times 64 \times 64$ was used as an input to the U-net model.

### 3.3. Preprocessing

Prior to vessel enhancement, Hounsfield unit (HU) windowing and Gaussian smoothing were used as a preprocessing step.

### 3.3.1. HU Windowing

Hounsfield unit(HU) windowing/intensity clipping is the contrast enhancement technique that highlights the particular structure. By adjusting the attenuation level and range in the image, the tissue of interest can be highlighted [29]. To enhance the hepatic vessel structures (Figure 1b), we used a HU range from 80 to 220.

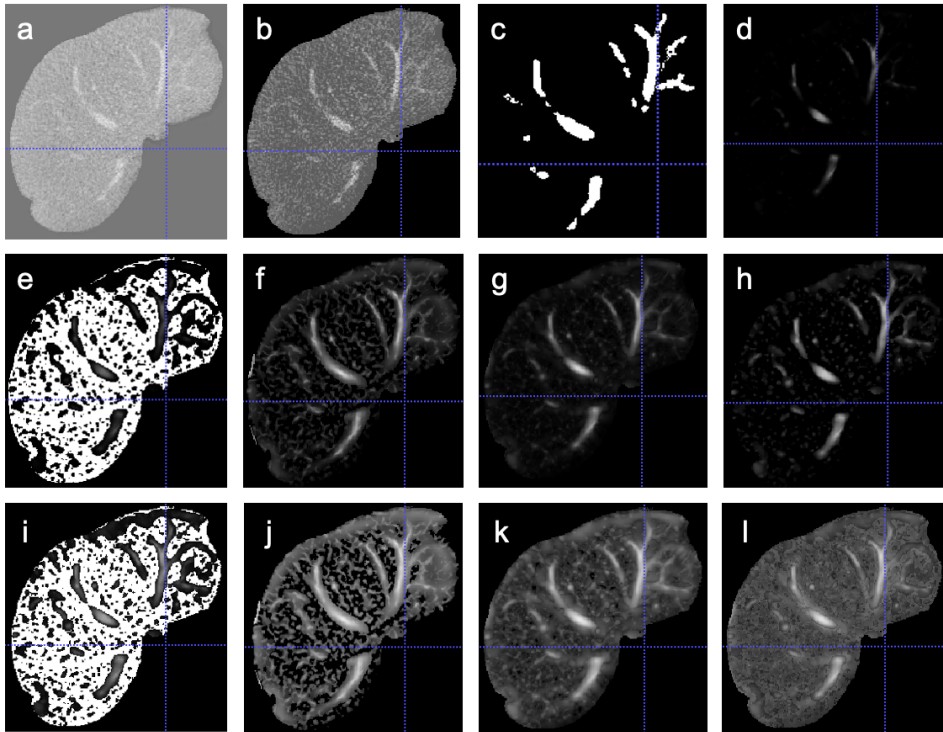

**Figure 1.** Comparison of different enhancement methods. (**a**) Original image, (**b**) HW, (**c**) Ground truth, (**d**) Frangi, (**e**) Hessian, (**f**) Meijering, (**g**) Sato, (**h**) FrangiGC, (**i**) HessianGC, (**j**) MeijeringGC, (**k**) SatoGC, (**l**) FilterAdded.

### 3.3.2. Gaussian Smoothing

Hessian-based vesselness filters are highly sensitive to noise and sharp edges. To reduce the noise, Gaussian smoothing or blurring of $\sigma = 1$ was applied prior to vessel enhancement filtering. Gaussian smoothing is given by,

$$G(x,y) = \frac{1}{2\pi\sigma^2} e^{-\frac{x^2+y^2}{2\sigma^2}} \tag{1}$$

### 3.4. Vesselness Enhancement

We investigated four different Hessian-based vessel enhancement filter techniques in our work. They are based on the second-order derivatives of the image intensity identifying the curvilinear structures in the image. These filtering techniques compute the Hessian matrix of the image f(X) where X = (x,y,z). The Hessian matrix is defined by

$$H(f) = \begin{bmatrix} h_{11} & h_{12} & h_{13} \\ h_{21} & h_{22} & h_{23} \\ h_{31} & h_{32} & h_{33} \end{bmatrix} = \begin{bmatrix} \frac{\partial^2 f}{\partial x_1^2} & \frac{\partial^2 f}{\partial x_1 \partial x_2} & \frac{\partial^2 f}{\partial x_1 \partial x_3} \\ \frac{\partial^2 f}{\partial x_2 \partial x_1} & \frac{\partial^2 f}{\partial x_2^2} & \frac{\partial^2 f}{\partial x_2 \partial x_3} \\ \frac{\partial^2 f}{\partial x_3 \partial x_1} & \frac{\partial^2 f}{\partial x_3 \partial x_2} & \frac{\partial^2 f}{\partial x_3^2} \end{bmatrix} \tag{2}$$

To reduce the effect of noise and to tune the filter to the width of the structures, the standard deviation $\sigma$ Gaussian kernel is applied to image I along with the second

derivative. This makes the filter, a multi-scale framework where the vessel scale depends on the $\sigma$.

$\lambda_1, \lambda_2, \lambda_3$ are the eigen values and $e_1, e_2, e_3$ are the corresponding eigenvectors of $H(f)$. On sorting $\lambda_1 \leq \lambda_2 \leq \lambda_3$, $e_1$ represents the direction along which the second derivative is maximum, i.e., the direction of the vessel. The eigenvectors $e_2$ and $e_3$ corresponds to the cross section directions of the vessel. This is referred to as the tube model [5,10]. Figure 1 shows an example image when different vesselness filtering techniques are applied.

### 3.4.1. Frangi Vesselness Filter

To discriminate between the different structures, the Frangi vesselness filter [10] used three eigenvalues. Based on these three eigenvalues, three measures to discriminate blobs ($R_b$), distinguish between the plate and line structures ($R_a$), Hessian norm measure to reduce the influence of noise ($S$) were derived.

$$R_b = |\lambda_1| / \sqrt{|\lambda_2\lambda_3|} \tag{3}$$

$$R_a = |\lambda_2||\lambda_3| \tag{4}$$

$$S = \sqrt{\lambda_1^2 + \lambda_2^2 + \lambda_3^2} \tag{5}$$

These measures are used in the function,

$$F = \left(1 - \exp\left(-\frac{R_a^2}{2\alpha^2}\right)\right) \exp\left(-\frac{R_b^2}{2\beta^2}\right)\left(1 - \exp\left(-\frac{S^2}{2c^2}\right)\right) \tag{6}$$

The parameters $\alpha$, $\beta$, c are the thresholds that control the function.

### 3.4.2. Hybrid Hessian or Hessian Vesselness Filter

Ng et al. [30] proposed a modified multi-scale Hessian filter by combining the directional gradient and Hessian matrix. The directional gradient of the image is computed and used to calculate the Hessian matrix H at specific scale $\sigma$. Each approximation in the H is the convolution between the directional gradient and Gaussian kernel. To obtain the texture orientation, the eigenvalues $\lambda_1$ and $\lambda_2$ are derived and used to compute the curve derivation R and similarity measure S. Using R and S, the curvilinear likeliness $\epsilon$ is derived and given by [30],

$$\mathcal{E}(x,y,\sigma) = \begin{cases} 0 & \text{if } \lambda_2 < 0 \\ e^{-\frac{\mathcal{R}}{2\beta_1^2}}\left[1 - e^{-\frac{s}{2\beta_2^2}}\right] & \text{otherwise} \end{cases} \tag{7}$$

The final output $\mathcal{L}$ of the filter will be the maximum of all scales that approximate the size of the ridges, given by

$$\mathcal{L}(x,y) = \max_{\sigma_{\min} \leqslant \sigma \leqslant \sigma_{\max}} [\mathcal{E}(x,y,\sigma)] \tag{8}$$

The parameters $\beta_1$, $\beta_2$ control the sensitivity of the filter.

### 3.4.3. Meijering Vesselness Filter

Meijering et al. [31] proposed a parameter-free vesselness function to detect elongated structures such as neurites. The method is based on the modified Hessian matrix $H'(f)$. The method was initially developed in 2D and later extended to 3D in [32], given by [16],

$$H'(f) = \begin{bmatrix} h_{11} + \frac{\alpha}{2}(h_{22} + h_{33}) & (1 - \frac{\alpha}{2})h_{12} & (1 - \frac{\alpha}{2})h_{13} \\ (1 - \frac{\alpha}{2})h_{21} & h_{22} + \frac{\alpha}{2}(h_{11} + h_{33}) & (1 - \frac{\alpha}{2})h_{23} \\ (1 - \frac{\alpha}{2})h_{31} & (1 - \frac{\alpha}{2})h_{32} & h_{33} + \frac{\alpha}{2}(h_{11} + h_{33}) \end{bmatrix} \tag{9}$$

In general, $\alpha = 1/3$. The eigenvalues of $H'(f)$ with respect to H(f) is expressed as

$$\lambda_i' = \lambda_i + \alpha\lambda_j + \alpha\lambda_k \tag{10}$$

for $i \neq j \neq k$.

The vesselness is defined by,

$$F = \begin{cases} \lambda_{\max}/\lambda_{\min} & \lambda_{\max} < 0 \\ 0 & \lambda_{\max} \geqslant 0 \end{cases} \tag{11}$$

where, $\lambda_{\max} = \max\{\lambda_1', \lambda_2', \lambda_3'\}$ which is computed at each voxel and $\lambda_{\min}$ is the minimum of all $\lambda_{\max}$ of the image.

### 3.4.4. Sato Vesselness Filter

Sato et al. [33] proposed a line enhancement filter function based on the tube model. It is given by,

$$F = \begin{cases} \lambda_c \exp\left(-\dfrac{\lambda_1^2}{2(\alpha_1\lambda_c)^2}\right) & \lambda_1 \leqslant 0, \lambda_c \neq 0 \\ \lambda_c \exp\left(-\dfrac{\lambda_1 2}{2(\alpha_2\lambda_c)^2}\right) & \lambda_1 > 0, \lambda_c \neq 0 \\ 0 & \lambda_c = 0 \end{cases} \tag{12}$$

where $\alpha_1 < \alpha_1$, $\lambda_c = \min\{-\lambda_2, -\lambda_3\}$. Sato *et al.* [33] sorted the eigenvalues $\lambda_i$ as $\lambda_1 \geq \lambda_2 \geq \lambda_3$ The parameters $\alpha_1$ and $\alpha_2$ control the assymmetrical strength and formulation.

### 3.5. Gamma Correction

Gamma correction or gamma ($\gamma$) is a non-linear operation used in histogram adjustment. The intensity of each pixel in the image scaled to the range [0,1] and raised to the power of $\gamma$ and scaled back. It is given by the expression [34],

$$V_{out} = AV_{in}^{\gamma} \tag{13}$$

where, A is a constant, $V_{in}$ is the original image and $V_{out}$ is the gamma-corrected image.

When $\gamma < 1$, the fainter objects becomes more intense while the brighter objects remain the same. If $\gamma > 1$, the medium intensity objects becomes fainter while the brighter objects remain the same.

### 3.6. Masking

To reduce the search space of the 3D U-net, the liver region of interest (ROI) is obtained using the U-net model [35,36]. The liver ROI does not include the inlet of the major liver vessels. So, the hepatic vessel ground truth labels were added to get the final liver mask (Figure 2). Hessian-based vesselness filters are very sensitive to gradient change. In order to avoid the false positives occurring in the liver borders, the liver is masked in the final step after the application of vessel enhancement filters.

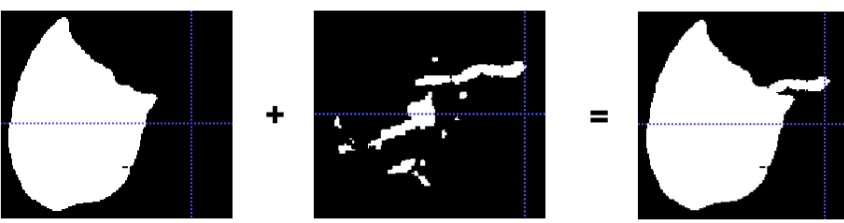

**Figure 2.** Masking. Liver ROI (**left**), vessel ground truth (middle), final liver mask (**right**).

### 3.7. Evaluation

In order to evaluate the results from the experiments, both quantitative and qualitative evaluations were performed. The quantitative evaluation is based on different evaluation metrics and the qualitative evaluation is based on the visual inspection and comparison analysis between predicted and ground truth segmentation.

#### 3.7.1. Quantitative Evaluation

In order to evaluate the segmentation performance, we have selected five different evaluation metrics. These include spatial overlap-based assessment methods such as DICE, Volume Overlap Error (VOE) and Relative Absolute Volume Difference (RAVD) and distance-based measures such as Average symmetric surface distance (ASSD) and maximum symmetric surface distance (MSSD). Below, we briefly describe these metrics.

#### 3.7.2. Dice Coefficient (DICE)

The Dice coefficient (DICE) is the most commonly used metric for validation in medical image segmentation. It is used to find the overlap between the ground-truth segmentation $S_g$ and the predicted segmentation $S_p$ using

$$DICE = \frac{2|S_g \cap S_p|}{|S_g| + |S_p|} \tag{14}$$

where $|S_g|$ and the $|S_p|$ are the cardinalities of the two sets.

#### 3.7.3. Volume Overlap Error (VOE)

Volume Overlap Error is defined as

$$VOE = 1 - \frac{V_{S_p} \cap V_{S_g}}{V_{S_p} \cup V_{S_g}} \tag{15}$$

where $V_{S_p}$ and $V_{S_g}$ are the volumes of segmented regions in the predicted segmentation and ground-truth segmentation, respectively. A value of zero for VOE corresponds to perfect segmentation.

#### 3.7.4. Relative Absolute Volume Difference (RAVD)

Relative Absolute Volume Difference is another volume-based metric and is defined as the ratio of the absolute difference between two segmentation volumes (predicted and ground-truth) and the volume of ground-truth segmentation

$$RAVD = \frac{|(V_{S_p} - V_{S_g})|}{V_{S_g}} \tag{16}$$

A value of RAVD closer to 0 generally corresponds to better segmentation, whereas a higher value suggests poorer segmentation.

#### 3.7.5. Average Symmetric Surface Distance (ASSD)

Average Symmetric Surface Distance is defined as the average of all the distances from boundary points of the segmented image regions to those in the ground-truth. It is given by the following equation:

$$ASSD(A, B) = \frac{\sum_{a \in A}[dist(a,b)] + \sum_{b \in B}[dist(b,a)]}{N_A + N_B} \tag{17}$$

where $A$ and $B$ represent the predicted and ground-truth surfaces, respectively, $N_A$ and $N_B$ are the number of points on each $A$ and $B$, and $dist(a, b)$ is the directed distance between mesh points on $A$ and $B$ and is given by

$$dist(a, b) = \min_{b \in B} ||a - b|| \tag{18}$$

with $||.||$ being the Euclidean distance. A smaller value of ASSD implies better segmentation results.

### 3.7.6. Maximum Symmetric Surface Distance (MSSD)

Maximum Symmetric Surface Distance is referred to as the maximum of two directed distances between the regions in the ground-truth and segmented image. Similar to ASSD, a smaller value of MSSD implies better segmentation results. For two finite point sets, MSSD is defined in terms of directed distance $dist(a, b)$ as

$$MSSD(A, B) = \max(\max_{a \in A}(dist(a, b)), \max_{b \in B}(dist(b, a))) \tag{19}$$

### 3.7.7. Qualitative Visual Evaluation

Qualitative visual inspection and evaluation were performed by the same medical doctor who made initial ground truth segmentations. To review predictions from the 3D U-net, 3D Slicer was used to load the original CT images and ground truth segmentations together with predictions. Firstly, the predictions generated from the different enhancement methods were visually inspected. The continuity of major vessels and main bifurcations, which have great clinical importance understanding inflow and outflow in the liver were mainly focused. Secondly, the under- and over-segmentations by calculating the difference between ground truth and predictions were created. These segmentations were visually inspected and used for HU calculation.

## 4. Experiments

The three experiments conducted in this work are presented in this section. In the first experiment, the effect of different vesselness filters were compared over unenhanced images. In the second experiment, the effects of gamma correction on vesselness filters were studied. In the third experiment, the effect of the fused/combined filter approach was studied.

### 4.1. Experiment 1: Comparison of Different Vesselness Filters

The aim of the first part of the experiment is to compare the results of the unenhanced hepatic vessel segmentation with the segmentation results obtained using different vesselness filters. The second part of the experiment is to compare Frangi, Sato, Meijering and Hessian vesselness filters. For this experiment, a total of five different models were trained. The first model was trained with the images without any vessel enhancement filter. Four other models were trained on Frangi, Sato, Meijering and Hessian vesselness filtered images (Figure 1d–g), respectively. The workflow of the experiment using these five models is shown in Figure 3.

### 4.2. Experiment 2: Effect of Gamma Correction on Vesselness Filters

In this experiment, the effect of the gamma correction technique on the four vesselness filters was studied. The gamma correction was applied on vesselness filtered images and these images (Figure 1h–k) were used as an input to the four 3D U-net models (Figure 4). The segmentation results obtained using gamma-corrected images were compared with the vesselness filtered images without gamma correction.

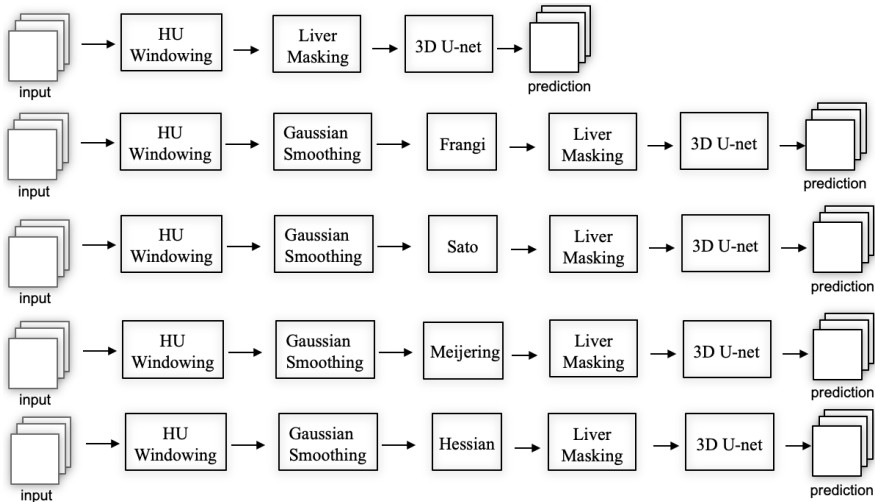

**Figure 3.** Workflow of the unenhanced and different vesselness enhancement approaches.

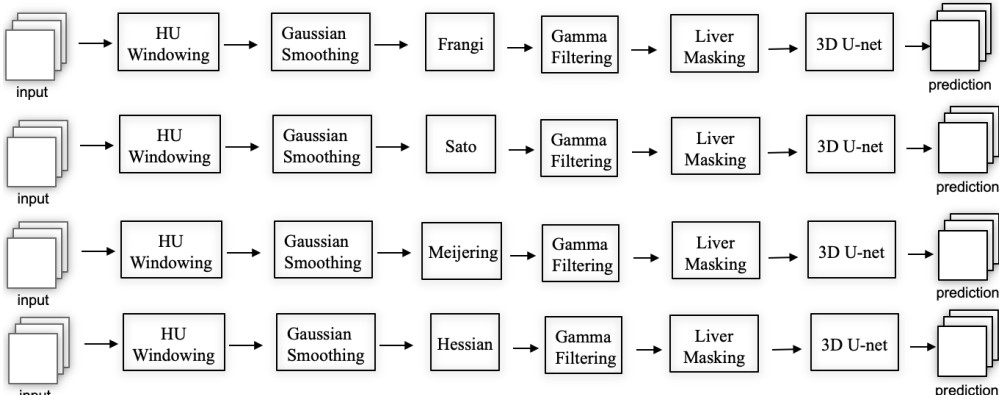

**Figure 4.** Workflow of different vesselness enhancement approaches with gamma correction.

### 4.3. Experiment 3: Fused Vesselness Filters

In experiment 3, we studied the effect of fused/combined vesselness filters on the individual filters. For this experiment, two different designs were studied. In the first design, the output from the four different filters followed by gamma correction was added channel-wise (Figure 1l). The U-net model was trained on the vesselness added image (FilterAdded (Figure 5)). In the second design, the output or the prediction from the four models trained using the four vesselness enhanced gamma-corrected images were added (SegAdded (Figure 6)) and analyzed. In order to reduce false positives, only the pixels predicted from at least two different filters were considered:

$$SegAdded = (F_g \cap H_g) \cup (F_g \cap M_g) \cup (F_g \cap S_g) \cup (S_g \cap H_g) \cup (S_g \cap M_g) \cup (H_g \cap M_g) \quad (20)$$

where $F_g$, $H_g$, $M_g$ and $S_g$ are Frangi-, Hessian-, Meijering- and Sato- gamma corrected images respectively.

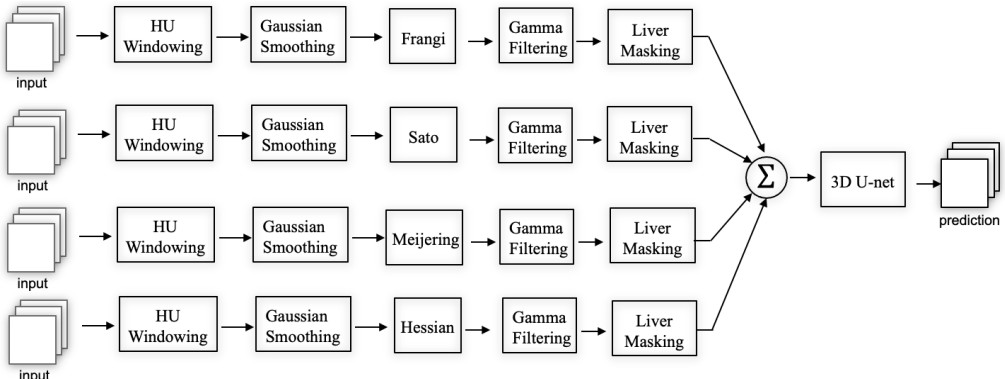

**Figure 5.** Workflow of the combined vesselness filter output (FilterAdded).

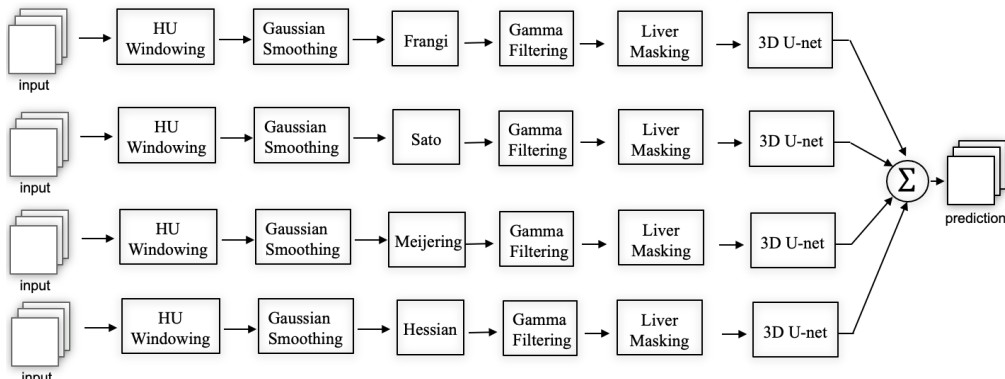

**Figure 6.** Workflow of the combined prediction from different vesselness filters (SegAdded).

## 5. Results

The results from the experiments based on both the technical evaluation and clinical evaluation are analyzed in this section.

### 5.1. Quantitative Evaluation

In order to evaluate the quality of segmentation, we used the five evaluation metrics presented in Section 3.7.1 namely DICE, VOE, RAVD, ASSD and MSSD. Figure 6 shows the boxplots for each of these metrics for Experiment 1, where the model trained using unenhanced images is compared to those trained with filtered images without Gamma correction. From the DICE boxplot in Figure 6a, we can observe that the models trained on Sato filtered and Meijering filtered give the best median values and overall spread whereas that based on Frangi gives the worst performance amongst the filtered images. A similar trend is also observed for VOE (Figure 6d). For ASSD and MSSD (Figure 6b,e), the median value of the model trained on unfiltered images is the best. This is followed in order by Meijering, Sato, Hessian and Frangi for ASSD and by Hessian, Sato, Frangi and Meijering for MSSD. However, in terms of consistency, all the filtered models can be considered to be superior to the model trained on unenhanced images on the basis of ASSD and MSSD with Hessian being the best, followed by Sato. Finally, for RAVD (Figure 6c), we also observe the best median value for unenhanced followed closely by Hessian, Meijering, Sato and Frangi. In terms of consistency, all models have a similar spread. However, it is important to note that for each metric unenhanced has an outlier value with a very high deviation which may be attributed to the lack of robustness of this model. Overall, Sato and Meijering can be considered to be the best performing model in Experiment 1 with Hessian as the second-best. This conclusion can also be confirmed on the basis of Table 1 where the mean and standard deviation of different metrics for each model are presented. From the table, we can see that, amongst the five models compared, Meijering has the best values for all metrics except ASSD, with Sato being the second-best for DICE and VOE,

indicating a good overlap between the two volumes. For distance-based metrics, Hessian gives the best mean values which are closely followed by Sato and Meijering. For RAVD, unenhanced has the best mean value, although the value of standard deviation is very high in this case.

**Table 1.** Mean and standard deviation values for Segmentation metrics with different filtering.

| Filtering | DICE | ASSD | RAVD | VOE | MSSD |
|---|---|---|---|---|---|
| Unenhanced | *0.740 ± 0.121* | *1.982 ± 2.135* | *0.310 ± 0.380* | 0.340 ± 0.136 | *3.132 ± 3.887* |
| Frangi | 0.764 ± 0.057 | 1.285 ± 0.294 | 0.229 ± 0.136 | *0.378 ± 0.073* | 1.683 ± 0.438 |
| Hessian | 0.782 ± 0.037 | 1.186 ± 0.172 | 0.198 ± 0.113 | 0.356 ± 0.050 | 1.481 ± 0.298 |
| Meijering | 0.800 ± 0.024 | 1.213 ± 0.289 | 0.197 ± 0.085 | 0.333 ± 0.033 | 1.624 ± 0.384 |
| Sato | 0.796 ± 0.032 | 1.202 ± 0.246 | 0.221 ± 0.119 | 0.338 ± 0.044 | 1.638 ± 0.491 |
| FrangiGC | 0.778 ± 0.045 | 1.236 ± 0.235 | 0.215 ± 0.131 | 0.360 ± 0.059 | 1.577 ± 0.426 |
| HessianGC | 0.802 ± 0.031 | 1.182 ± 0.254 | 0.156 ± 0.092 | 0.330 ± 0.042 | 1.462 ± 0.325 |
| MeijeringGC | 0.814 ± 0.026 | 1.153 ± 0.216 | **0.150 ± 0.081** | 0.313 ± 0.037 | 1.431 ± 0.288 |
| SatoGC | 0.814 ± 0.026 | 1.022 ± 0.175 | 0.179 ± 0.085 | 0.313 ± 0.037 | 1.294 ± 0.226 |
| SegAdded | **0.830 ± 0.026** | **0.979 ± 0.191** | **0.150 ± 0.084** | **0.290 ± 0.039** | 1.227 ± 0.269 |
| FilterAdded | **0.818 ± 0.031** | **0.995 ± 0.223** | 0.159 ± 0.076 | **0.306 ± 0.043** | **1.206 ± 0.235** |

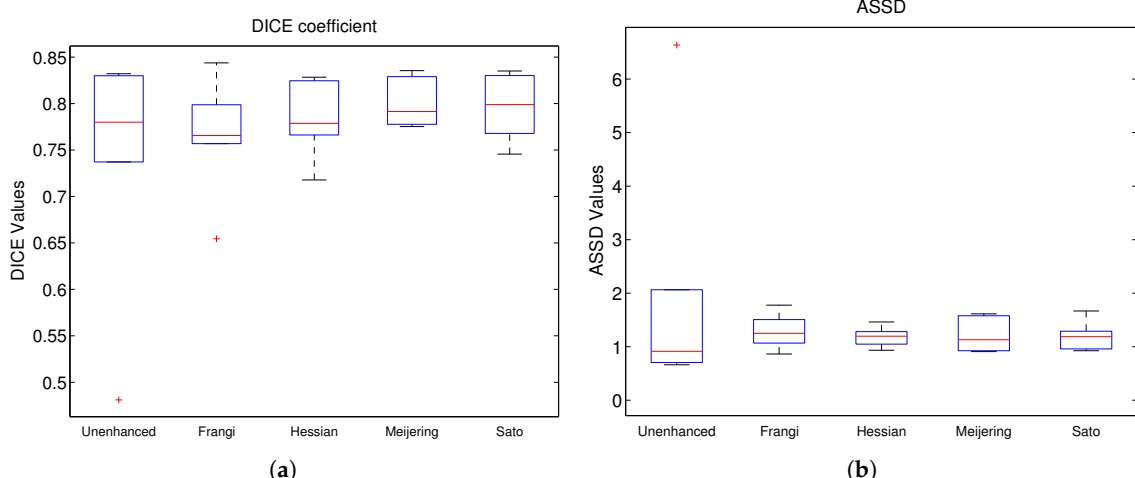

(**a**)          (**b**)

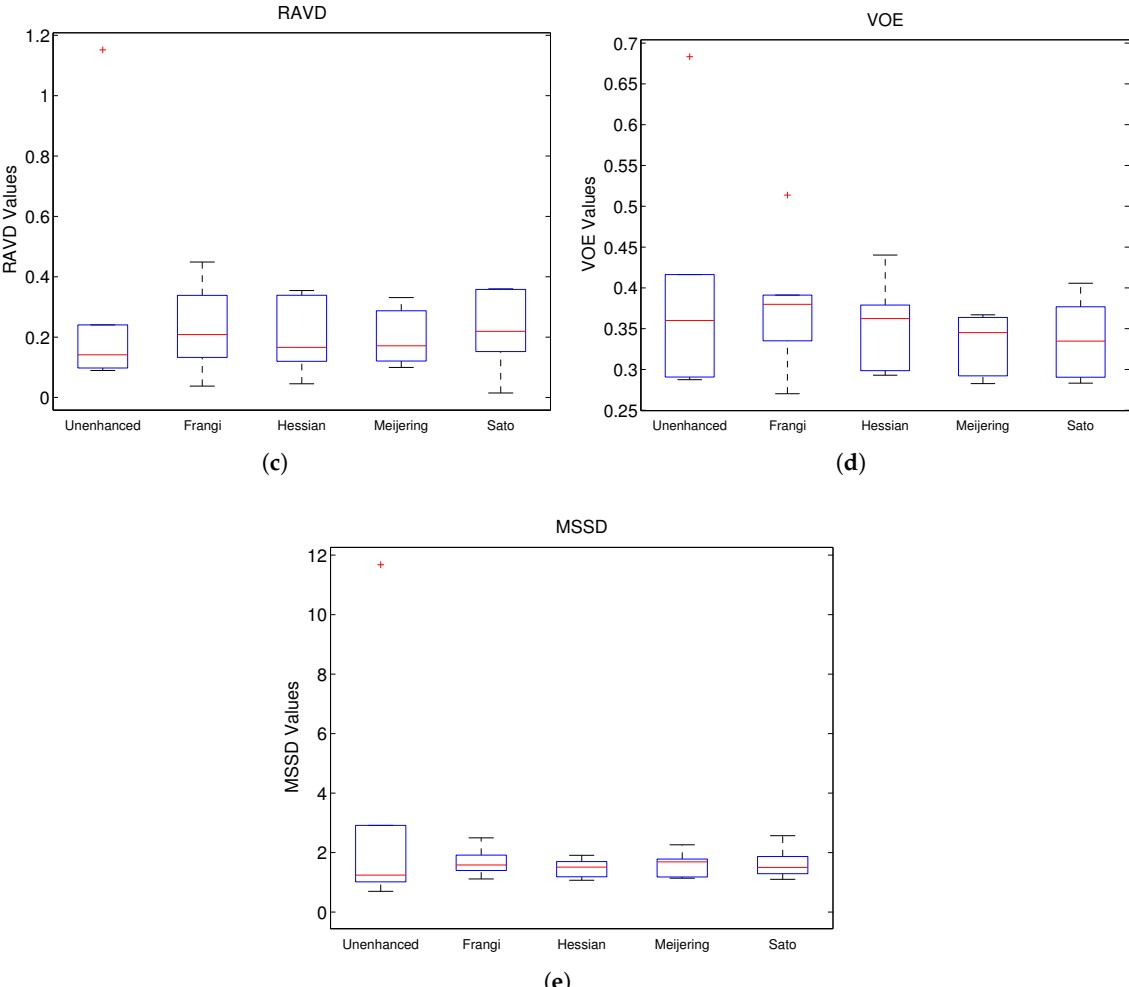

**Figure 6.** Comparison of segmentation for unenhanced and different filtered images (red + are outliers). (**a**) DICE; (**b**) ASSD; (**c**) RAVD; (**d**) VOE; (**e**) MSSD.

In Experiment 2, we have also included the results of models trained on filtered and gamma-corrected images. Figure 7 shows the results from this experiment. The left-most box in each boxplot in the figure is that from the unenhanced whereas the remaining boxes are presented in pairs to highlight the differences between the models trained on filtered images with and without gamma correction. In each pair, the boxes in blue represent results from models without gamma correction whereas those in red correspond to those with gamma correction. Overall, we can observe that the models trained with gamma-corrected versions perform much better than their corresponding counterparts with better median values and similar spreads. From among all the methods compared SatoGC has, in general, the best values for all the five metrics followed by MeijeringGC. On the other hand, unenhanced and Frangi give the worst performance from all the methods compared. This holds true also for comparison between mean and standard deviation values as can be seen in Table 1.

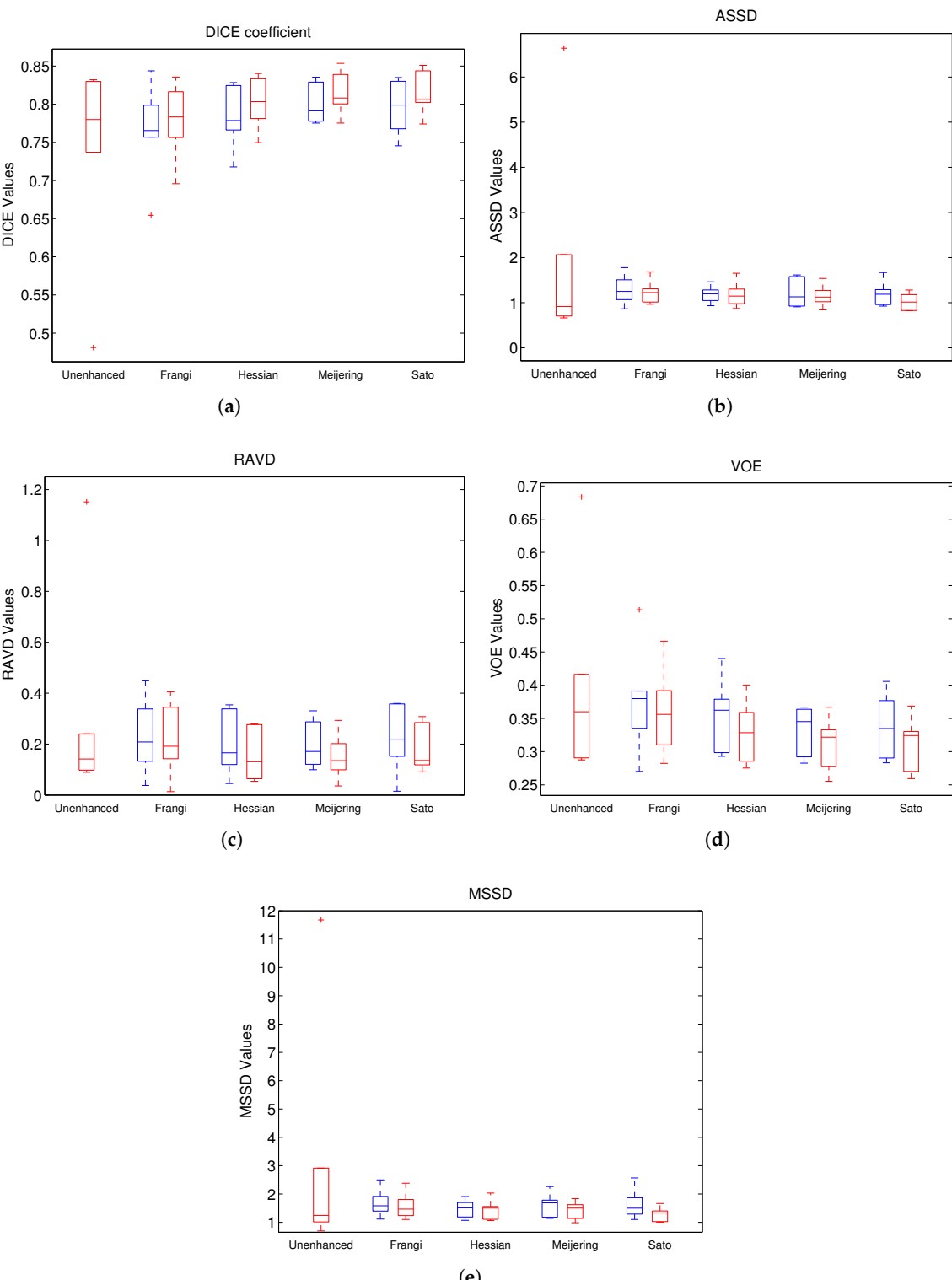

**Figure 7.** Comparison of segmentation of unenhanced (first red) and filtered images without (blue) and with (red) Gamma correction. (**a**) DICE; (**b**) ASSD; (**c**) RAVD; (**d**) VOE; (**e**) MSSD.

Finally, for Experiment 3, we have compared the results of the two configurations in Figures 5 and 6 denoted by SegAdded and FilterAdded with unenhanced. These results are shown in Figure 7. From the results, we can clearly observe that both these methods outperform unenhanced in terms of median values and overall consistency. Both have higher DICE values and lower values for all other metrics depicting better perfor-

mance. If we compare the performance of the two methods with each other, FilterAdded is the better one with the best median values and a smaller spread for all the metrics. To get an idea of the overall best- and worst-performing methods, we have plotted boxplots of all the methods from the three experiments in Figure 7. For DICE (Figure 7a), FilterAdded has the highest median value with a small spreads and hence is the best performing method. This is followed by SegAdded and SatoGC methods. Among the worst-performing are the unenhanced and Frangi. For ASSD (Figure 7b), the best three are the same but Frangi is the worst in terms of median, whereas unenhanced has the lowest consistency despite a better median value. The same holds true for the MSSD metric (Figure 7e). Similarly, for RAVD (Figure 7c) and VOE (Figure 7d) the best three are FilterAdded, SegAdded and SatoGC, respectively, whereas Frangi is the worst for RAVD and unenhanced is the worst for VOE with a very large spread. Table 1, displays the mean and standard deviation values from all the filtered and unfiltered models. The values highlighted in bold indicate the best performance for each metric, whereas those in italic are the worst. From the table, we can observe that, as expected, most of the conclusions made based on the boxplots hold true in terms of these values also. In terms of mean and standard deviation values, SegAdded shows the best performance overall with FilterAdded not very far behind. To analyze the significance, we also performed paired t-tests to compare different filtering methods with the two best methods and found improvement using SegAdded to be statistically significant from all of them for majority of metrics. The same results were also found for comparison of all filtered methods with FilterAdded, with the only exception of MeijeringGC and SatoGC, where only values from one of the five metrics were found to be statistically more significant. Hence, in terms of these quantitative results, it can be considered safe to conclude that the SegAdded and the FilterAdded methods are the best whereas the unenhanced and the Frangi give the poorest segmentation results. Figure 8 shows the comparison between both the Frangi and SegAdded which has the worst and best mean Dice score respectively.

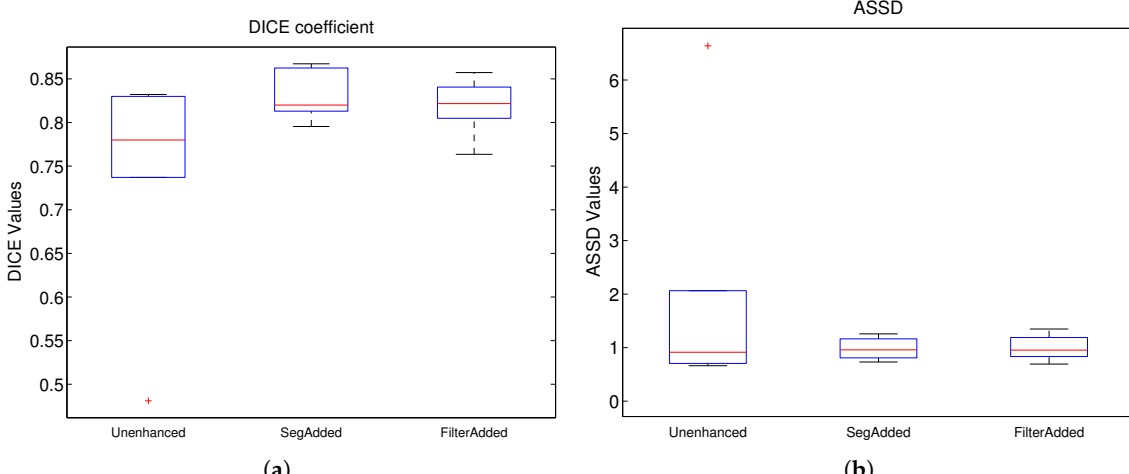

(**a**)                                                                                                  (**b**)

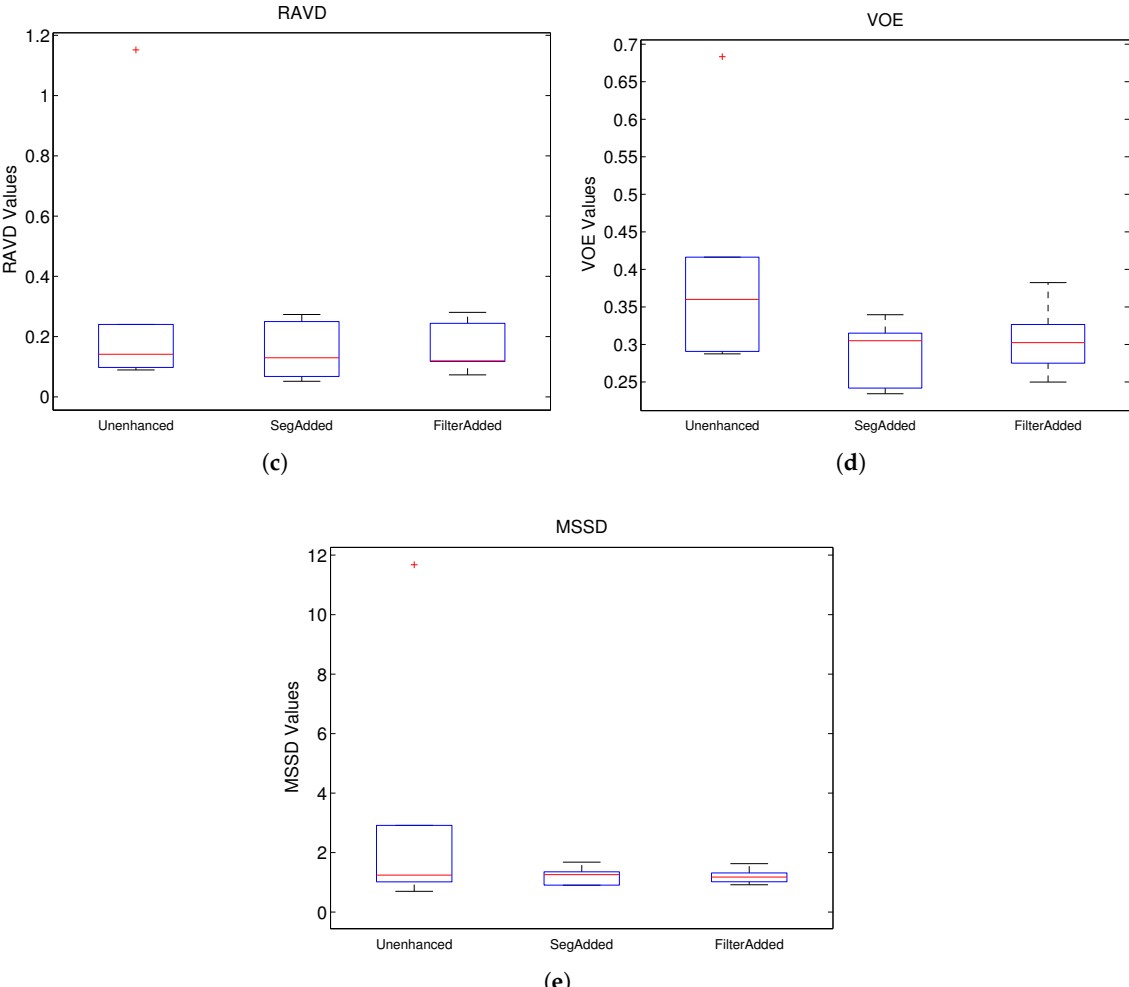

**Figure 7.** Comparison of segmentation of filtered images with combination of filters. (**a**) DICE; (**b**) ASSD; (**c**) RAVD; (**d**) VOE; (**e**) MSSD.

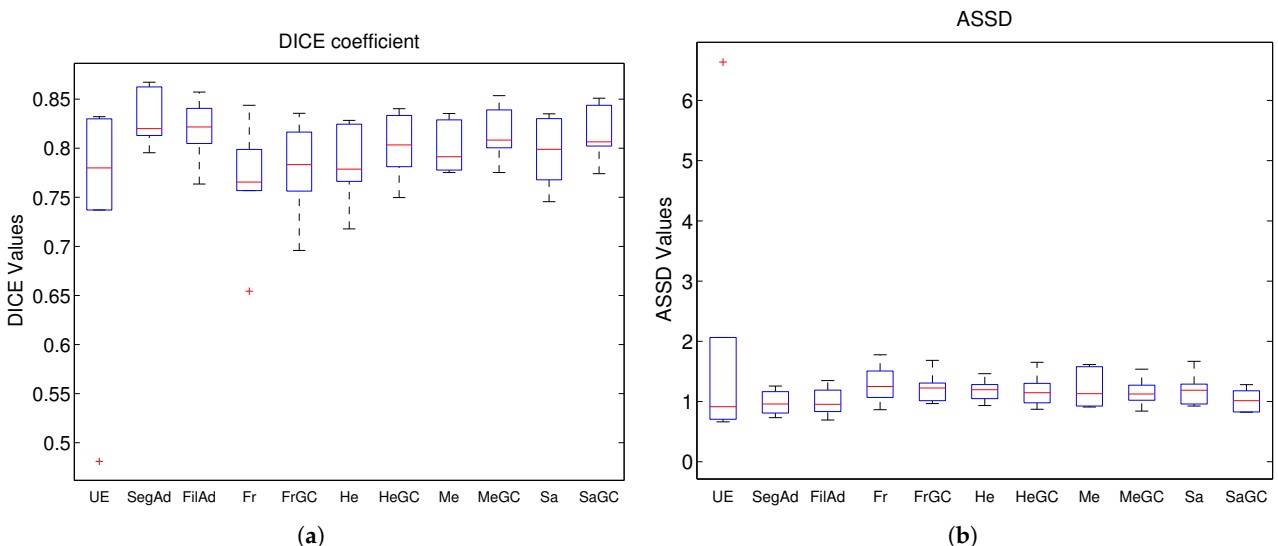

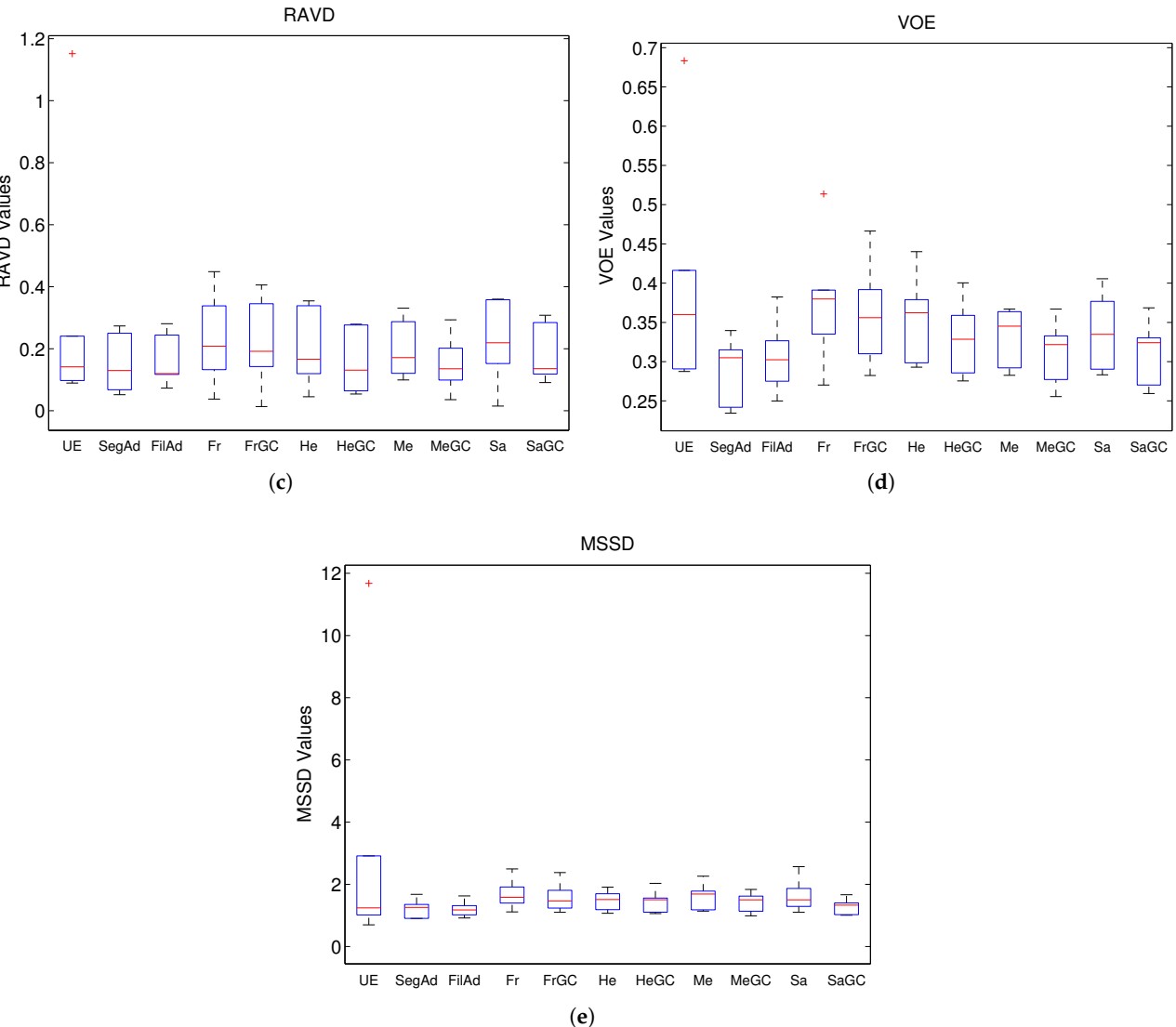

**Figure 7.** Comparison of all the methods. UE—Unenhanced, SegAd—SegAdded, FilAd—FilterAdded, Fr—Frangi, FrGC—FrangiGC, He—Hessian, HeGC—HessianGC, Me—Meijering, MeGC—MeijeringGC, Sa—Sato, SaGC—SatoGC. (**a**) DICE; (**b**) ASSD; (**c**) RAVD; (**d**) VOE; (**e**) MSSD.

### 5.2. Qualitative Evaluation

General inspection of the predictions show that results are good as an initial segmentation of the vessels which can be manually corrected, though different fusions and disruptions of vessels are present, examples shown in Figure 9. These errors are clearly seen by visual inspection of the inferences together with calculated territories of over- and under-segmentation with examples shown in Figure 10. Most of the errors are in the edges and borders of vessels. This is supported by values in the CT images showing lower average HU for both over- and under-segmentation, shown in Table 2.

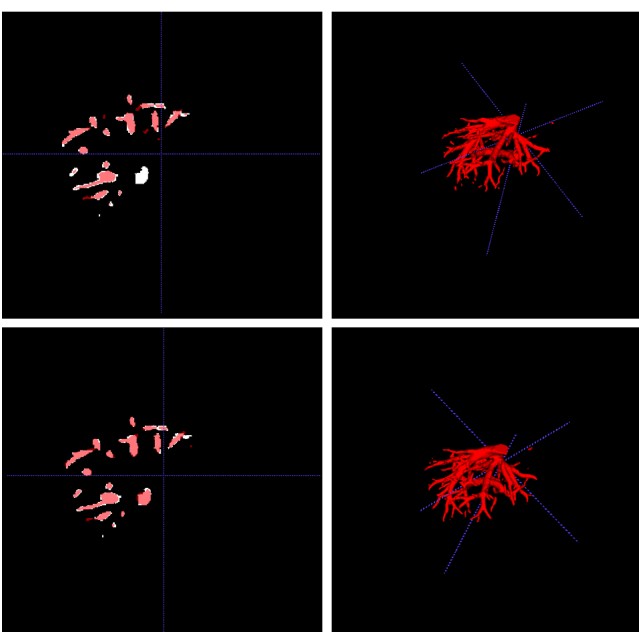

**Figure 8.** Liver vessel segmentation overlay on ground truth using Frangi enhanced image which has the worst Dice score; 2D slice (top-left) and 3D view (top-right), and SegAdded image which has the best Dice score; 2D slice (bottom-left) and 3D view (bottom-right).

**Table 2.** Average overall HU for test data evaluation with over- and under-segmentation.

| Label | Overall | Over-segmentation | Under-segmentation |
|---|---|---|---|
| Ground truth | $151.91 \pm 33.30$ | N/A | N/A |
| Unenchanced | $149.60 \pm 38.50$ | $119.89 \pm 40.32$ | $135.60 \pm 31.90$ |
| Frangi | $157.23 \pm 32.16$ | $123.16 \pm 26.28$ | $131.86 \pm 29.12$ |
| Hessian | $155.28 \pm 33.89$ | $123.25 \pm 25.83$ | $133.91 \pm 26.31$ |
| MeijeringGC | $153.08 \pm 33.50$ | $124.05 \pm 25.90$ | $132.81 \pm 27.73$ |
| SatoGC | $154.57 \pm 33.32$ | $125.42 \pm 25.17$ | $132.40 \pm 26.16$ |
| FrangiGC | $157.04 \pm 32.03$ | $124.71 \pm 26.05$ | $131.32 \pm 29.05$ |
| HessianGC | $153.78 \pm 33.94$ | $123.96 \pm 25.11$ | $133.67 \pm 25.95$ |
| Meijering | $154.77 \pm 33.40$ | $124.13 \pm 26.36$ | $133.05 \pm 27.60$ |
| Sato | $156.17 \pm 33.66$ | $128.31 \pm 25.39$ | $129.84 \pm 26.40$ |
| SegAdded | $153.85 \pm 33.29$ | $126.06 \pm 24.82$ | $131.61 \pm 25.96$ |
| FilterAdded | $154.04 \pm 33.59$ | $125.05 \pm 25.25$ | $132.41 \pm 25.84$ |

Overall, the predictions from the deep learning model had an average HU $124.36 \pm 26.95$ for over-segmentation and $132.59 \pm 27.45$ for under-segmentation compared to an average $151.91 \pm 33.30$ for the ground truth. Both over- and under-segmentation capture HU values on average lower than ground truth which indicates that most of the errors are in the edges or additional territories with lower amounts of contrast. In some cases and some slices, predictions in several filtered images captured vessels which were not segmented in the ground truth, examples shown in Figure 9.

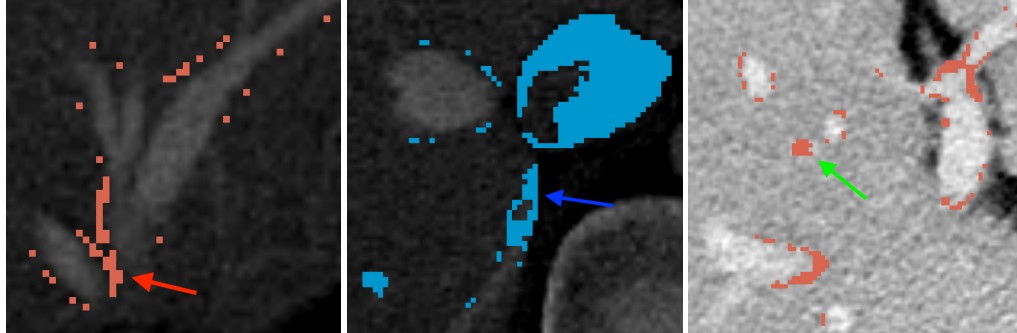

**Figure 9.** Left image—an example of oversegmentation in the borders of the vessels with a vessel fusion in the bifurcation (red arrow). Middle image—undersegmentation disrupting vessel continuity (blue arrow). Right image—an example of a segmented vessel which was not detected and segmented in the ground truth (green arrow).

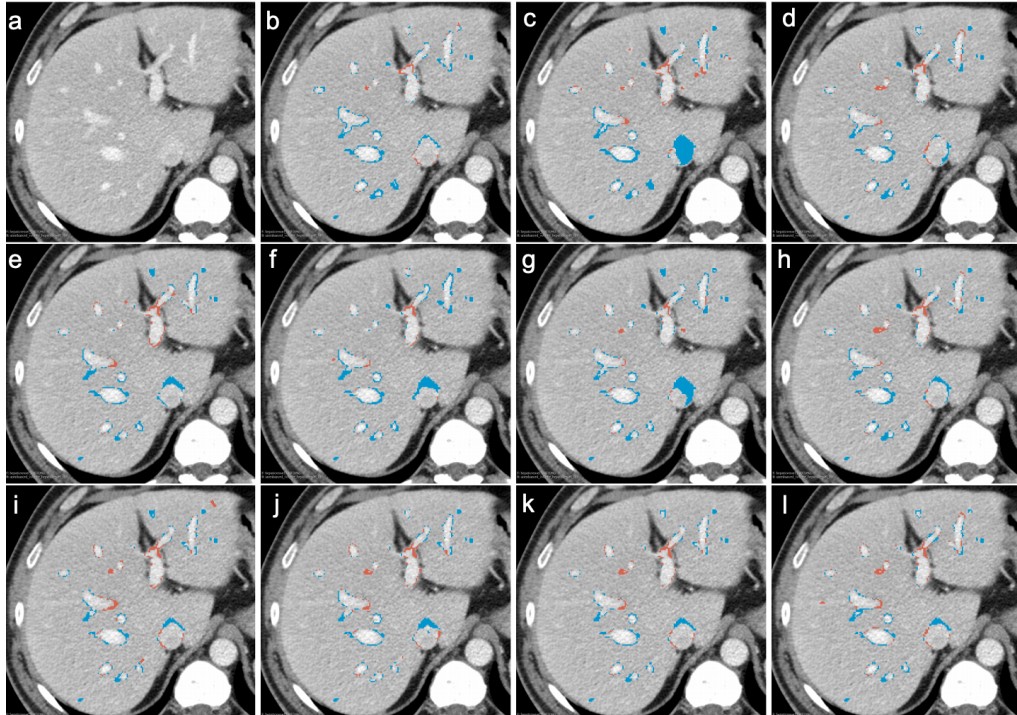

**Figure 10.** A slice in the CT images from test data followed by examples of oversegmentation (shown in red) and undersegmentation (shown in blue) from all image enchanchements tested. (**a**) Original image, (**b**) Unenhanced, (**c**) Frangi, (**d**) Hessian, (**e**) Meijering, (**f**) Sato, (**g**) FrangiGC, (**h**) HessianGC, (**i**) MeijeringGC, (**j**) SatoGC, (**k**) SegAdded, (**l**) FilterAdded.

The model trained on unenhanced images generated a visually more "noisy" segmentation with more variations and more extreme errors, shown in Figure 11. Additionally, the model trained on the unenhanced images had the lowest values of over-segmentation and the highest values of under-segmentation. This can be a sign that these predictions include more of the background in the over-segmentation and misses more high-intensity zones in the under-segmentation.

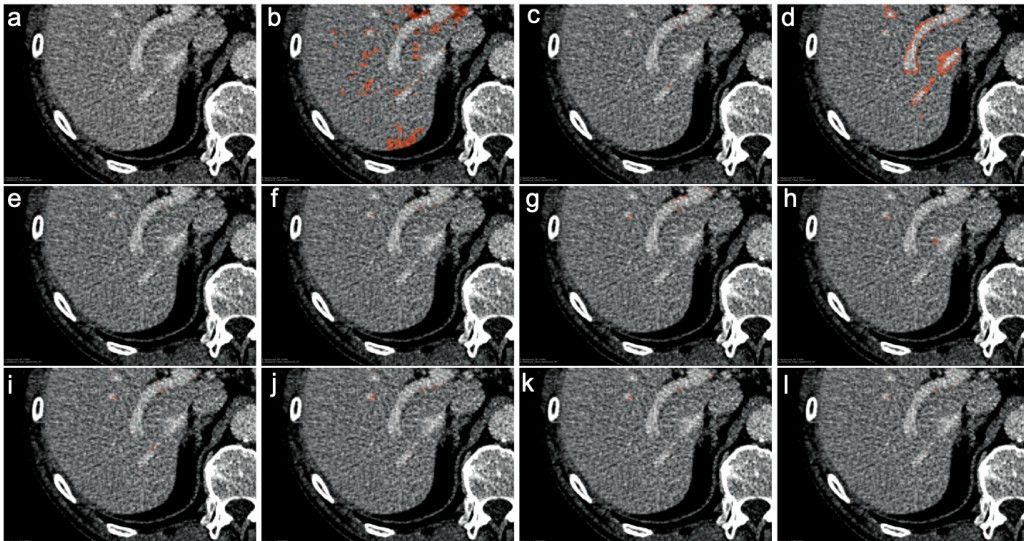

**Figure 11.** A slice in the CT images from test data followed by examples of over-segmentation (shown in red) from all image enhancements tested. (**a**) Original image, (**b**) Unenhanced, (**c**) Frangi, (**d**) Hessian, (**e**) Meijering, (**f**) Sato, (**g**) FrangiGC, (**h**) HessianGC, (**i**) MeijeringGC, (**j**) SatoGC, (**k**) SegAdded, (**l**) FilterAdded.

Overall, visually the SegAdded and the FilterAdded provided a more consistent segmentation of the liver vessels, compared to the other tested methods, and had sometimes advantage of capturing finer details.

## 6. Discussion

In experiment 1, on comparing the effect of Frangi, Hessian, Sato and Meijering vesselness enhancement filters with the unenhanced images, the model trained on Sato enhanced images seem to perform the best and the model trained on Frangi enhanced images has the worst performance. In addition to the quantitative evaluation, from the Figure 1, we see that the Sato and Meijering enhanced images seem to capture the vessel information better compared to Hessian and Frangi enhancement. Especially when applying the Frangi vesselness filter, we noticed there is a huge loss of vessel information due to which, the model trained on these images has poorer performance than the model trained on unenhanced images.

In experiment 2, on comparing the effect of gamma correction on Frangi, Hessian, Sato and Meijering vesselness images, we see that the models trained on gamma-corrected images perform better than the ones without gamma correction. Due to gamma correction, the fainter vessel structures in the enhanced images were further enhanced and the models trained on the gamma-corrected enhanced images seem to capture the vessel structure better than the images without gamma correction.

In the final experiment, we studied the effect of fusing all four enhancement filters with gamma correction. In FilterAdded, the FrangiGC, HessianGC, SatoGC and Meijering GC were fused and the model was trained on the fused image. In SegAdded, four different models were trained on FrangiGC, HessianGC, SatoGC and MeijeringGC individually and the segmentation maps from these four models were combined to produce the final segmentation. On comparing FilterAdded and SegAdded, we observed very similar performances with FilterAdded, giving better overall median values and SegAdded giving better mean values. However, the differences between the mean and median metric values were not found to be significant.

On combining the results from all the experiments, the FilterAdded and SegAdded from experiment 3 have the best performance followed by SatoGc. The overall improvement in the performance was found to be statistically significant in terms of the majority of the evaluation metrics for most cases and in terms of at least one metric for Meijer-

ingGC and SatoGC. From the results and the statistical test, we observe that the fused methods have better potential to capture the vessel structures compared to the individual enhancement filters.

Qualitative visual inspection of different results from the methods tested indicates high variability in the vessel edges for all the approaches. Visually, SegAdded and FilterAdded had more consistent predictions with various differences in fine detail depending on the test case. In some slices, vessels were predicted and afterward confirmed by the clinician although defined as oversegmentation because they were not included in the original ground truth. It is important to mention that manual segmentation is a subjective task and is highly user dependent. Therefore ground truth segmentation should not be defined as perfect or complete because there is room for different interpretations and improvements.

Improvement to automatic segmentation might be achieved with the addition of more quality data to train the deep learning model. This work and method provide a solid starting point for vessel segmentation, which can be further worked on using manual correction tools.

## 7. Conclusions

Automatic segmentation of hepatic vessels is critical for computer-assisted liver surgery, treatment planning and navigation. For the segmentation of complex structures such as liver veins, enhancing the vessel structures makes the segmentation tasks less challenging. In this work, the effects of four different vesselness filters with and without gamma correction have been studied. Additionally, the effect of fused vesselness enhancement over individual filters has been studied. The quantitative analysis of the results in terms of different evaluation metrics from experiments shows that each of the filtered methods improves the segmentation results as compared to those that are unenhanced. Moreover, it was observed that, by applying gamma correction, a statistically significant improvement was achieved in the performance of each filter with SatoGC and MeijeringGC giving better results. Finally, our study showed that both the fused filtered images and fused segmentation give the best results in terms of all the five evaluation metrics with a statistically significant improvement compared to the individual filters with and without Gamma correction. The worst performance was observed for that which was unenhanced and for a model with Frangi (without gamma correction). In addition to that, qualitative evaluation of the deep learning-based segmentations showcase current pitfalls and potential to be used clinically although after extensive manual corrections. To conclude, this work provides an important contribution towards the improvement of the outcomes of the challenging hepatic vessel segmentation task, by making intelligent use of the existing vesselness filters for enhancement in combination with deep learning methods.

In the future, we would like to extend the study to clinical MRI volumes and cross-modality vessel enhancement studies for multi-label vessel segmentation (portal and hepatic vein separated).

**Author Contributions:** Conceptualization, S.S. and F.L.; methodology, S.S., R.P.K. and F.L.; software, S.S.; validation, S.S., Z.A.K. and E.P.; formal analysis, Z.A.K. and E.P.; labeled dataset creation, E.P.; writing—original draft preparation, S.S., Z.A.K. and E.P.; writing—review and editing, S.S., Z.A.K., E.P., F.L., R.P.K., and B.E.; supervision, F.L. and R.P.K.; project administration and funding acquisition F.L. All authors have read and agreed to the published version of the manuscript.

**Funding:** This work is supported by H2020-MSCA-ITN Marie Skłodowska-Curie Actions, Innovative Training Networks (ITN) -H2020 MSCA ITN 2016 GA EU project number 722068 High Performance Soft Tissue Navigation (HiPerNav).

**Institutional Review Board Statement:** The study was conducted according to the guidelines of the Declaration of Helsinki, and approved by the Local and Regional Review Board. This study utilizes a dataset derived from Oslo University hospital OSLO-COMET (Oslo Randomized Laparoscopic Versus Open Liver Resection for Colorectal Metastases Trial). (ClinicalTrials.gov: NCT01516710).

**Informed Consent Statement:** Informed consent was obtained from all subjects involved in the OSLO-COMET trial.

**Data Availability Statement:** Restrictions apply. The data are not publicly available due to privacy or ethical issues.

**Conflicts of Interest:** The authors declare no conflicts of interest.

## Abbreviations

The following abbreviations are used in this manuscript:

| | |
|---|---|
| CRC | Colorectal Cancer |
| CNN | Convolutional Neural Networks |
| CLAHE | Contrast Limited Adaptive Histogram Equalization |
| RORPO | Ranking the Orientation Response of Path Operators |
| CT | Computer Tomography |
| MRI | Magnetic Resonance Imaging |
| MRA | Magnetic Resonance Angiography |
| CTA | Computed Tomography Angiography |
| HU | Hounsfield unit |
| FrangiGC | Frangi enhanced image with Gamma Correction |
| SatoGC | Sato enhanced image with Gamma Correction |
| HessianGC | Hessian enhanced image with Gamma Correction |
| MeijeringGC | Meijering enhanced image with Gamma Correction |
| FilterAdded | Filtered images Added |
| SegAdded | Segmentation maps Added |
| ROI | Region of Interest |
| DICE | Dice Coefficient |
| VOE | Volume Overlap Error |
| RAVD | Relative Absolute Volume Difference |
| ASSD | Average Symmetric Surface Distance |
| MSSD | Maximum Symmetric Surface Distance |

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
