# Peer review of "Effects of Enhancement on Deep Learning Based Hepatic Vessel Segmentation"

_electronics, doi:10.3390/electronics10101165_

Round 1
Reviewer 1 Report
The experiments were well structured resulting in a well-written paper. It is true, that the positive merits of "just right" gamma correction for improving image recognition are well known. The same goes for improving on a model by mixing all possible algorithms, or as it is in this case, using all possible filters, to get "a couple of points more".
For the qualitative analysis, you mention:
"In some slices, vessels were predicted and afterward confirmed by the
clinician although defined as oversegmentation because they were not included in the original ground truth. It is important to mention that manual segmentation is a subjective task and is highly user dependent. Therefore ground truth segmentation should not be defined as perfect or complete because there is room for different interpretations and improvements."
Was the original ground truth defined only by one person or more? Afterwards, were the new vessels confirmed by one clinician or more and was/were those the same person/people as for the original ground truth? The qualitative results would carry more weight if independent QC would be stated.
All in all, the authors did take it upon them to test out and combine the usual suspects and find a solution then provides a significant improvement compared to simple filtering. Good job!
P.S. Just a small note...In the first sentence, checking your first reference, in 2020, colorectal cancer came to projected 3rd place (after breast and lung), with a total of cca. 1,9 million new cases (male + female).
Reviewer 2 Report
The paper is well written in all its parts
The paper talks about the effects of Enancement on Deep Learning Based Hepatic Vessel. In particular the paper tells that a precise knowledge of the patient's anatomy and liver pathology plays a fundamental role in the treatment process. To have it often it is necessary to use the segmentation process to create 3D models. Segmentation of the hepatic vessels is challenging due to the large variations in the size and directions of the vessel structures. The work focuses on the combination of different vesselness enhancement filters and preprocessing methods to enhance the hepatic vessels prior to segmentation, using 3D-U-net, a widely Deep Learning based segmentation model in the medical imaging domain. Finally a quantitative and qualitative analysis was made.
Suggestion:
In the figure 3, Figure 4, Figure 5 and Figure 6 I suggest to extend the description of the figure in the Figure label. In particular you could write the experiments with a detailed descriptions.
In Figure 7, Figure 8, Figure 9 the y-axis label has not been inserted.Without this information it could be difficult to interpret the graphs.It may be useful to insert a label in the y axis to better understand the graphs.
Figure 14 is too far to the left of the other figures.To standardize the formatting of the paper, move the figure to the right along with the figure label.
Reviewer 3 Report
Summary: The authors have applied various image enhancement techniques to images of livers. 3D U-net is used to segment the vessels. The results compare the various enhancement methods.
Comments:
In my opinion, the introduction could be expanded to include a summary of the presented method/work.
The related work section could be improved by setting the scene (how does the cited work relate to your work) and mentioning how your approach or/and application domain differs from the related works.
The experiments/results could be expanded: Do the observed trends hold true when other image segmentation approaches are used?
Minor comments:
line 98: "work and The" should be changed to "work. The".
Line 145: “i.e” -> “i.e.”
Grammar could be improved.
In the first paragraph of Section 3.3, mention that Figure 1 shows an example image for when different preprocessing techniques are applied.
In Figure 7, there are some red + marks, are these outliers? (Mention what they are within the text or caption.)
Rather than writing “fourth section” you could reference the section numbers, e.g., “In Section4...”.
